# Needs assessment for creation of a platform trial network in metabolic-dysfunction associated steatohepatitis
Elena Sena [1], Frank Tacke[2], Quentin M. Anstee [3,4], Nicholas Di Prospero[5], Mette Skalshøi- Kjær[6], Sergio Muñoz-Martínez[1], Jesús Rivera-Esteban[1], Alba Jiménez-Masip[1], Jesús M. Bañales[7], María Martínez-Gómez[1], Franz Koenig[8], Joan Genescà [1], Vlad Ratziu[9] & Juan M. Pericàs [1] ✉

## Abstract

**Background** The EU Patient-cEntric clinicAl tRial pLatforms (EU-PEARL) project (IMI2-853966) aimed to develop tools to establish integrated research platforms (IRP) for conducting adaptive-design trials in various diseases, including metabolic-dysfunction associated steatohepatitis (MASH). One essential component of a successful MASH IRP is a robust and reliable Clinical Research Network (CRN). Herein, we outline the required elements and anticipated steps to set-up such a CRN.

**Methods** We identified European clinical research sites that could potentially serve as the foundation for MASH IRP and a CRN. A survey was sent to sites to assess their interest in joining a CRN, their familiarity with platform trials, and their capacity to participate in a future MASH IRP.

**Results** A total of 141 investigators were invited to participate in the survey, and 40% responded. More than half of the answers (52%) identify MASH with advanced fibrosis (F3-4) as the subpopulation with the greatest unmet need. Regarding the difficulty in identifying candidates for trials, 65% find it is moderately difficult and 30% very difficult. Most respondents (94%) believe that a platform trial could offer substantial benefits to patients. Nearly all researchers express interest in participating in a platform trial (78%), with 22% indicating their interest would be contingent on initial industry funding.

**Conclusion** While preliminary, our findings on responding sites are encouraging for the potential establishment of a CRN for a MASH IRP. However, funding schemes and sustainability strategies to provide proof-of-platform in MASH seem key in the short-term scenario.

## Plain language summary

Metabolic dysfunction-associated steatohepatitis (MASH) occurs when the liver becomes damaged due to the build up of fat, which is often related to obesity and diabetes. There is a lack of effective drug treatments for MASH, so strategies to strengthen clinical research in this area are needed. Here, we survey key European experts on MASH to assess their interest in joining a network of MASH researchers and their interest in participating in a new type of clinical trial called a platform trial, where multiple drugs can be tested simultaneously. Researchers largely agree that these are promising approaches to boost drug development in the field, although have concerns regarding funding and sustainability strategies. Our findings may inform the creation of a network of MASH researchers capable of running a platform trial, which in turn may speed up research into treatments for MASH.

Metabolic dysfunction-associated steatotic liver disease (MASLD) and metabolic dysfunction-associated steatohepatitis (MASH)[1] have emerged as significant public health concern on a global scale[2]. MASLD is characterized by fat accumulation in the liver in individuals without significant alcohol consumption, while MASH represents a more severe form of MASLD, marked by inflammation, cellular damage, and fat accumulation in the liver[3]. Recent estimations indicate that MASLD and MASH affect up to 25-30% and 5-8% of the global population, respectively[4–6]. The growing burden

[1]Liver Unit, Vall d'Hebron University Hospital, Vall d'Hebron Institute for Research (VHIR), Universitat Autònoma de Barcelona, Spanish Network of Biomedical Research on Digestive and Liver Diseases (CIBERehd), Barcelona, Spain. [2]Department of Hepatology and Gastroenterology, Charité - Universitätsmedizin Berlin, Berlin, Germany. [3]Liver Unit, The Newcastle upon Tyne Hospitals NHS Foundation Trust, Newcastle upon Tyne, UK. [4]Translational and Clinical Research Institute, Newcastle University, Newcastle upon Tyne, UK. [5]Janssen Research and Development, Raritan, NJ, USA. [6]Novo Nordisk A/S, Bagsvaerd, Denmark. [7]Department of Liver and Gastrointestinal Diseases, Biodonostia Health Research Institute-Donostia University Hospital, University of the Basque Country (UPV/EHU), San Sebastian, Spanish Network of Biomedical Research on Digestive and Liver Diseases (CIBERehd), IKERBASQUE, Basque Foundation for Science, Bilbao, Spain. [8]Center for Medical Data Science, Medical University of Vienna, Vienna, Austria. [9]Department of Hepatology, Pitié-Salpetriere Hospital, University Paris 6, Paris, France. ✉e-mail: juanmanuel.pericas@vallhebron.cat

of MASLD/MASH is closely associated with the increasing rates of obesity and type 2 diabetes, both recognized as significant risk factors for MASLD[7]. MASH can lead to severe health complications, including cirrhosis, liver cancer, and liver failure[8]. Nevertheless, no regulatory-approved pharmacological treatments are currently available, largely due to challenges associated with conventional standalone trials in MASH, especially the assessment of combined histologic endpoints based on liver biopsy (Table 1)[9].

During the EU IMI2 Patient-cEntric clinical tRial pLatforms (EU-PEARL) project, we have established the groundwork for an integrated research platform (IRP) designed to conduct platform trials (PTs) as an alternative approach to expedite drug development in the MASH field[9–11]. A PT differs from a conventional randomized clinical trial in that a patient can be assigned to a variety of drugs or interventions that are simultaneously examined through different subtrials within the platform, with the possibility of sharing a common placebo arm[12,13]. The advantages of a PT include the capacity to test multiple therapies simultaneously, which can accelerate the process of developing new treatments. Furthermore, these trials can enhance efficiency and cost-effectiveness compared to traditional clinical trials by enabling the sharing of resources and data among various therapies under investigation and by the standardization of protocols and interpretation[12].

A key element of IRPs are clinical research networks (CRN), which are designed to facilitate clinical research by providing infrastructure, coordinating mechanisms and resources necessary for the efficient deployment of a PT through an IRP (Fig. 1). While there are existing initiatives aimed to foster research collaborations among MASH experts (e.g., the EU IMI2 Liver Investigation: Testing Marker Utility in Steatohepatitis (LITMUS) consortium[14] and the associated European NAFLD Registry[15]), a CRN focused on operationalizing a PT in MASH is currently lacking. With this aim, during the EU-PEARL journey, we elaborated a series of materials outlining the characteristics of the CRN for a future IRP in MASH[16]. This future CRN should foster a multidisciplinary collaboration amongst MASH researchers, featuring a centralized and shared infrastructure that includes databases, biobanks, and other resources to facilitate collaboration, data sharing, and analysis among network participants as well as common cores for radiology, pathology, and statistics for centers, along with protocols for sample handling, distribution and management. Moreover, the CRN should include multiple institutions across diverse regions to ensure diversity in the patient population and enhance patient engagement. Finally, a robust system for data management, quality control, and monitoring should be in place to ensure data integrity and patient safety throughout the trial. To ensure the realization of these characteristics, active engagement of all the stakeholders (e.g., researchers, drug owners and patients) is essential within both the CRN and the IRP.

The role of several parties involved in a CRN is largely determined by the IRP's sponsorship. Due to intellectual property aspects, most PT are sponsored by either academic institutions or non-profit organizations. We hypothesize that this is also the case for MASH. There is an additional reason why the initial proof-of-concept for a PT in the MASH field (i.e., proof-of-

platform) will largely rely on academic institutions, namely because, as there is not yet a regulatory-approved drug in the market, companies are focused on a high-stakes race to market a novel drug, so they are less open to cross-company collaboration[10] (Table 2). Hence the interest to gather information about the potential involvement of academic researchers in participating in a CRN to build an IRP for MASH.

Herein, we present the results of an initial reach-out survey that was sent to European sites to assess their potential interest in participating in a MASH IRP and collect their input, and we reflect on how the survey's results and EU-PEARL's conclusions overall might shape the next steps to set up a PT for MASH. Respondents identify patients with advanced liver fibrosis as having the most significant therapeutic unmet need in the MASH population and also express their concerns about the difficulty of identifying candidates for MASH trials and the reliance on liver biopsy as a diagnostic tool. The most valued advantage of participating in a platform trial for the respondents is the increased chances for patients to receive an active compound. All surveyed sites are willing to participate in a future platform trial, with some expressing interest in industry-funded trials or research consortiums for funding the MASH IRP. The statistical considerations and overall design rationale for the master protocol to be applied during the platform trial can be found elsewhere[17,18], as well as the governance aspects, the business plan and feasibility of the platform trial[19].

## Methods
### Interest survey
To set up the foundation of a CRN for the MASH IRP, the EU-PEARL MASH taskforce members contacted potential EU and UK clinical research sites candidates to form the core. A survey was sent in December 2022 to the selected sites Supplementary Information File (Supplementary Note 1) in order to assess their capabilities and interest in becoming part of a CRN as well as their knowledge about PT and their ability and readiness to participate in a future MASH IRP.

### Identification of sites
To create a list of potential sites that could become the MASH IRP CRN, sites were identified as being capable of recruiting patients for a MASH PT based on public data on clinicaltrials.gov[20], and centers associated with the European NAFLD Registry[15]. Moreover, additional sites were added through secondary sources, e.g., recommendations by other respondents of the survey. Most sites included in the list are recognized as experienced sites with the capacity to recruit MASH patients and possess the technical capabilities to perform any necessary procedures.

A graphical summary of the survey process is shown in the Supplementary Information File (Supplementary Fig. 1).

### Scope of the survey
In addition to assessing the readiness of sites to conduct clinical trials in MASH and therefore their feasibility as potential recruiting sites within a MASH IRP, we expanded the scope to gather information on other relevant areas. These include respondents' awareness regarding PTs and their

## Table 1 | Unmet needs in MASH drug development and potential role of a platform trial

| | |
|---|---|
| Unmet needs in MASH drug development | • Long-term trials requiring assessments of both histological endpoints and clinical outcomes.<br>• Lack of adoption of non-invasive biomarkers to diagnose and monitor the disease by regulatory agencies.<br>• Unwillingness of participants to either undergo liver biopsies and/or receive placebo.<br>• Means to minimize screening failures.<br>• Some ethnicities underrepresented and several special populations not included in clinical trials (e.g. people living with HIV, children, women with childbearing potential). |
| Potential role for a platform trial | • Reducing logistical complexity.<br>• Enhancing recruitment and allocation to subtrials within the platform.<br>• Use of non-invasive tests for endpoints and outcomes.<br>• Evaluation of more than one drug in the same trial, lowering screening failure rates.<br>• Patients have more chances to receive active treatment.<br>• Increasing efficiency by selecting those compounds or combinations with higher likelihoods to achieve endpoints and therefore be graduated to the following phase based on interim analyses (Bayesian forecasting).<br>• Incorporation of additional compounds or combinations at any time when they become available. |

peculiarities, feedback on the master protocol we developed to carry out a Phase 2b PT in MASH[18], the potential interest in participating in a future MASH IRP and their willingness to be involved according to the funding sources of such IRP (e.g., whether they would be willing to be part of research consortium to apply for European Commission grants or just would be open to participate as recruiting sites once funding is secured). The full survey can be consulted in the Supplementary Information File (Supplementary Note 1).

## Data collection and ethics
Study data were collected and managed using REDCap electronic data capture tools hosted at Vall d'Hebron Institute of Research[21,22]. The questions were designed to explore the resources and ability of sites to recruit patients for MASH clinical trials but, more importantly, to assess their interest in being part of a CRN for the MASH IRP. Vall d'Hebron University Hospital Campus IRB approved the study protocol (code PR(AG)461/2021) and participants provided their informed consent when responding the survey through REDCap.

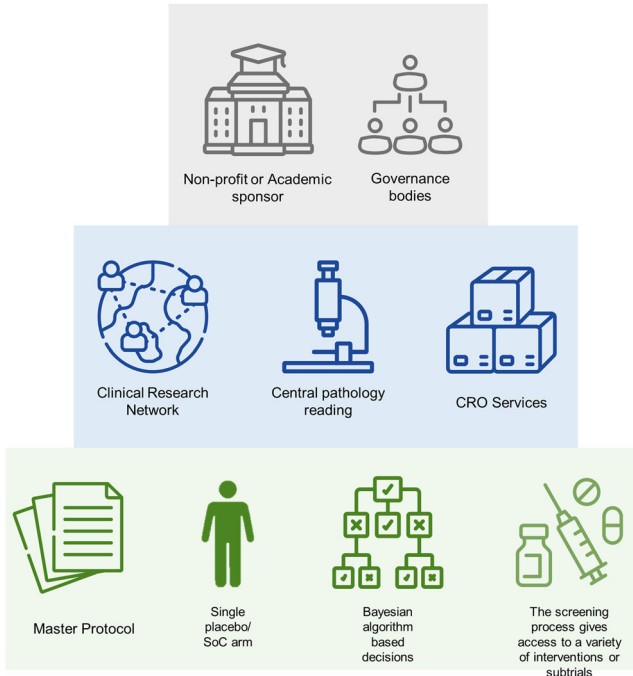

**Fig. 1 | Integrated research platform (IRP) elements.** Different elements and stakeholders are key to the functioning of an IRP. In the figure, elements involving the development of a master protocol, the governance bodies of the IRP and some logistical elements are depicted. IRP integrated research platform.

The steps and timelines of the survey are summarized in Supplementary Information File (Supplementary Fig. 1). The complete survey is available in Supplementary Information File (Supplementary Note 1).

## Statistical analysis
The answers to the survey were expressed as absolute frequencies and percentages (%). The survey was developed using the REDCap electronic data capture tool. Raw data and results were directly extracted from the platform.

## Reporting summary
Further information on research design is available in the Nature Portfolio Reporting Summary linked to this article.

## Results
### Respondents' profile
The list of pre-selected sites contains 141 entries with a representation of most of European locations (see Supplementary Data 1). In some locations, more than one investigator was contacted. Some countries are more broadly represented (i.e., Spain, France, UK, Germany and Italy) while in other countries the contact information or sites availability is lower (i.e., Czech Republic, Hungary, Norway and Poland). Figure 2 and Table 3 illustrate and detail the number of contacted and participating sites by country. From the contacted investigators, 40% replied to the survey. By countries, Spain and Italy have the highest response rate. Most respondents (93%) have appointments at University Hospitals and 68% participate in either national or international registries for MASLD or MASH (see details on the characteristics of respondent sites in Supplementary Data 2).

### Patients' needs and engagement in MASH clinical trials
More than half of respondents (52%) state that the MASH population with a more significant therapeutic unmet need are the patients with advanced liver fibrosis, defined as a F3–F4 fibrosis (Fig. 3a). When asked about trial's participants, 65% of respondents declare that it is moderately difficult to identify candidates for MASH trials (Fig. 3b). Moreover, various respondents express their concern regarding the lack of non-invasive tests and the use of liver biopsy both as diagnostic gold standard and requirement in clinical trials from Phase 2b onwards. When asked about their opinion of the barriers for patients to be willing or able to participate in a MASH trial, there is a clear consensus around the liver biopsy supporting primary endpoints leads to a high proportion of potential trial candidates refusing to participate as well as screen failures. When asked about the possibility of having regulatory approval for MASH drugs not based on histology findings, 72% indicate that this will occur sooner or later. In addition, 38% of respondents answer that the combination of industry with academic institutions are most likely to lead the change, followed by academic institutions alone, in how MASH trials are conducted (Fig. 3c).

## Table 2 | Future scenarios for an integrated research platform to carry out adaptive trials in MASH

| | |
|---|---|
| Opportunities | • Multi-stakeholder collaboration: aligning patients, researchers and industry needs.<br>• Centralized and shared infrastructure: databases, biobanks and other resources.<br>• Enhanced logistics and compound graduation through interim analysis and platform adaptation.<br>• Coexistence of monotherapy and combination therapy subtrials within the platform.<br>• Global scope and representation of Ethnic minorities and special subpopulations.<br>• Boost patient engagement and embed patient-centered outcomes and experiences.<br>• Advance the field in terms of knowledge on the natural history of the disease and non-invasive biomarkers, e.g., cirrhotic population. |
| Challenges | • Logistical management: need for a strong CRO to back up procedures.<br>• Obtaining funding from drug owners.<br>• Regulatory landscape remains conservative. Thus, designing trials that are not based on liver biopsy-related endpoints might be risky.<br>• Proof-of-platform required at a small scale before some major stakeholders adopt adoptive designs as a preferred drug development strategy and leave behind the standalone trial as the only conceivable option.<br>• The culture of adaptive trials in general and platform trials in particular is still nascent in liver disease.<br>• An academic-led trial in a handful of countries could be the most suitable starting point for implementation. However, thus far only COVID-19 platform trials have been largely funded by public entities, whereas in other fields (e.g., oncology and hematology) private companies engagement has been pivotal. |

**Fig. 2 | Countries contacted for the survey.** Europe map showing countries with pre-selected sites to be invited to participating in the survey. Colors indicate the number of sites contacted per country based on the information available.

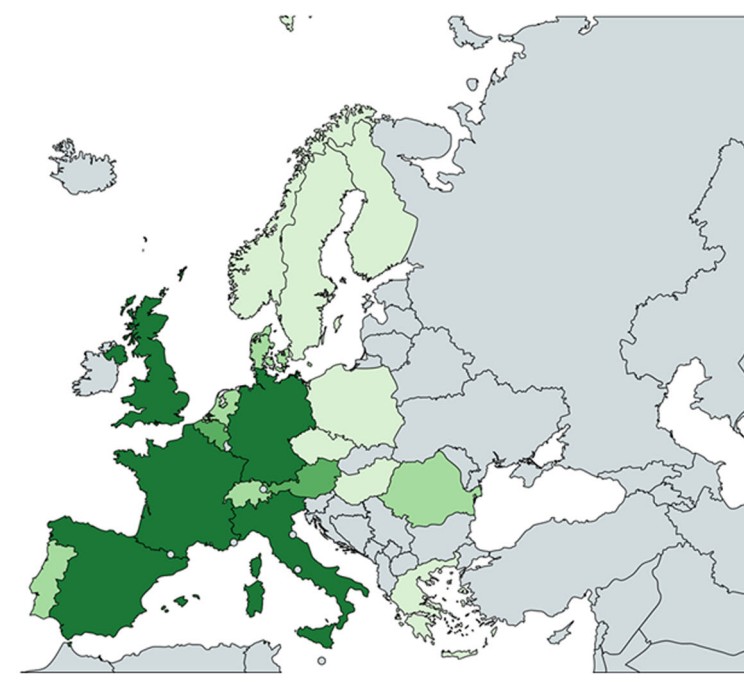

### Table 3 | Ratio of sites' response to the survey

|  | Invitations sent (*n*) | Total answers per country (*n* [%]) |
|---|---|---|
| Total | 141 | 57 (40 %) |
| Czech Republic | 1 | 0 (0%) |
| Hungary | 1 | 0 (0%) |
| Norway | 1 | 0 (0%) |
| Poland | 1 | 0 (0%) |
| France | 16 | 2 (13%) |
| Belgium | 7 | 1 (14%) |
| Austria | 6 | 1 (17%) |
| Switzerland | 5 | 1 (20%) |
| Romania | 3 | 1 (33%) |
| UK | 16 | 6 (38%) |
| The Netherlands | 5 | 2 (40%) |
| Germany | 14 | 6 (43%) |
| Spain | 36 | 16 (44%) |
| Finland | 2 | 1 (50%) |
| Greece | 2 | 1 (50%) |
| Denmark | 5 | 3 (60%) |
| Portugal | 5 | 3 (60%) |
| Italy | 12 | 10 (83%) |
| Sweden | 3 | 3 (100%) |

The table shows the total number of sites globally and per country and the number of answers expressed in percentage in reference to the total number of answers.

### Platform trials in MASH: experience, benefits and constraints

Seemingly, most respondents (58%) are familiar with adaptive trial designs. However, only 21% have ever participated in a PT. When asked whether the specific characteristics of the MASH landscape make it likely to arise interest from funders and regulators to set up a PT, 98% respondents answer positively. All respondents indicate that they would feel comfortable participating in a clinical trial that allows interim decisions to stop an arm due to futility/safety reasons. None of the respondents state that PT offer no extra benefits to patients compared to traditional standalone trials, only 6% of respondents believe that PT offer few extra benefits, whereas the rest of the respondents consider that PT provide either moderate or many benefits to patients (Fig. 3d). When asked about the concurrence with ongoing traditional clinical trials on MASH, 80% of respondents answer that they do not foresee any issue regarding feasibility or recruitment ability due to competing purposes. Those who state that there might be recruitment issues express that it is not particular for the PT but for any trial.

### Preferred characteristics of a potential platform trial in MASH

When asked to classify which item was more relevant to participate in a PT, the most valued advantage is the increased chances for patients to receive an active compound, followed by the scientific endeavor and innovation potential of the project. When asked about the type of sponsor for the MASH IRP, 85% of respondents prefer having an academic sponsor for the MASH PT, as well as a CRO specialized in MASH. Although more researchers prefer having a global CRN, 46% respondents agree with a European network. 54% of respondents state that they are more confident working with a large pharmaceutical company rather than with a small biotech, although in most of the cases (31%) the respondents choose the two options indistinctly. 94% of respondents state that they would be open to master protocol modification that replaces liver biopsy with non-invasive biomarkers as the primary endpoint. However, more than half of the respondents (57%) disagree with adapting the current protocol to a Phase 2a to avoid biopsy from the beginning. The ones that would agree mentioned benefits for patients and a faster development of drugs approval by doing it. Regarding the digitalization of biopsies, 100% agree that it should be like this in the PT and finally, 94% of respondents believe that the master protocol should be adapted in the future to encompass the cirrhotic population.

### Interest in participating in a MASH IRP

All surveyed sites are willing to participate in a future PT. Interestingly, 21% would be willing to participate only if the trial is funded by industry from the beginning. Those who are interested in participating, answer in a following question that they would be keen in participating in a research consortium aimed at gathering funding to provide proof-of-platform for establishing the MASH IRP (94%).

**Fig. 3 | Graph depicting answers to the following questions from the survey. a** Currently, what do you think is the MASH population with a more significant therapeutic unmet need?; **b** How difficult would you say it is to identify candidates for MASH trials?; **c** What type of stakeholder is more likely to take the initiative to change the way MASH trials are conducted?; **d** Do you believe that a platform trial can provide distinct benefits to MASH patients? All results are expressed in percentages. MASH metabolic dysfunction-associated steatohepatitis.

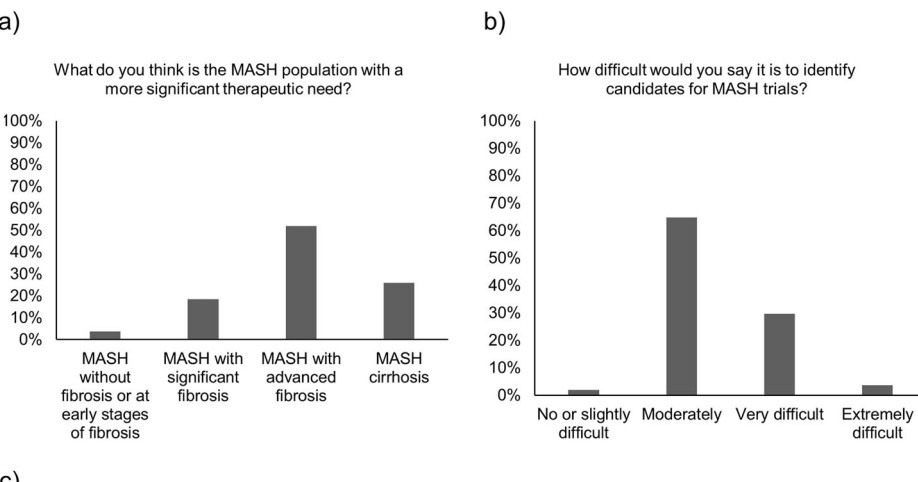

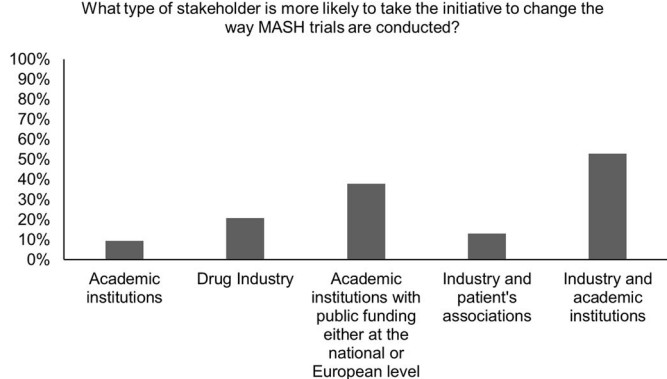

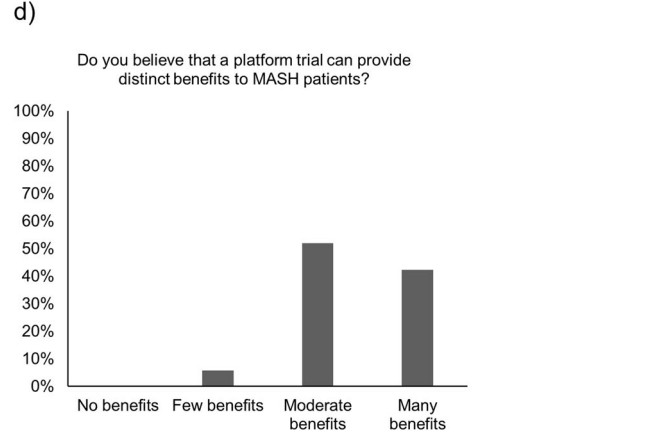

## Discussion

### Towards a platform trial in MASH: strengths and weaknesses

Setting up a PT is a multi-step, multilayered, multistakeholder process that requires at least five basic components: first, an unmet need; second, a robust scientific rationale; third, the interest of all parties involved; four, strong leadership and resilience; and lastly, the appropriate economic and human resources. The fulfillment of the first two requirements is evident in the case of MASH. However, there are *pros* and *cons* regarding the other three elements[9]. The results of the survey support the preliminary conclusions drawn during the development of EU-PEARL on the interest of the main actors in being involved in a MASH PT. There seems to be ample support and interest in participating in such trial by academic stakeholders in Europe, many of whom consider the involvement of companies owning compounds to treat MASH in their pipelines essential for the IRP success.

### Elements shaping the way forward for the setup of a platform trial in MASH

The establishment of a PT in MASH is influenced by several key determinants that play a crucial role in shaping its development and implementation. Current regulatory practices and recommendations are likely the most encompassing factor affecting the MASH drug development process as a whole. Even though there are increasing data on the efficacy of non-invasive tests to stage and monitor the progression of the disease, both steatohepatitis and fibrosis[23–27], current FDA/EMA recommendations rely on liver biopsy for non-cirrhotic patients and on clinical events or liver biopsy for the cirrhotic population[28].

The population of interest is another crucial determinant in the setup of a PT in MASH. Although the master protocol developed during EU-PEARL was based on a Phase 2b design for non-cirrhotic patients (fibrosis stages 2 and 3)[18], relying on histologic endpoints and therefore requiring

liver biopsies and the corresponding logistical complexity, at present it appears that focusing on cirrhotic patients could be a promising alternative. The main reasons are two: drug owners have started focusing their attention on cirrhotic patients and several trials on MASH cirrhosis have been launched in the last two years, and using non-invasive endpoints, including clinical events, is more feasible in the cirrhotic population. Thus, either academic or public-private endeavors could be envisioned in setting up an IRP on cirrhotic patients, perhaps using ordinal endpoints[25]. The survey results discussed in this report indicate that a significant proportion of investigators express interest in incorporating cirrhotic patients in the IRP, highlighting the importance of focusing on this population subset. More than half of the respondents acknowledged that the MASH population with the greatest therapeutic unmet need consisted of patients with advanced liver fibrosis. Respondents also highlighted the limitations associated with the need for liver biopsy and the exclusion and inclusion criteria for enrolling patients in MASH trials. These factors were considered significant barriers to patient recruitment. Additionally, an overwhelming 94% of respondents envisioned changes in the regulations concerning the use of NITs for diagnosing and monitoring the progression of MASH. This indicates a growing recognition of the potential benefits and utility of NITs in clinical practice and research.

The latter point is tightly related to the need of determining what the best approach to provide proof-of-concept is for a PT (proof-of-platform) in MASH. Different funding models can support the setup of a PT in MASH[8,9]. Yet, in order to persuade companies, it seems necessary to generating evidence showing that IRPs are efficient and patient-centric tools, also in the case of MASH. Although nearly 20% of respondents stated that in order to join the MASH IRP, they would expect industry funding to be already in place, one likely scenario is that a small-scale (one or few countries involved) academically-driven trial, perhaps focusing on repurposing drugs already available in the market with indications other than MASH is set up to gather such critical preliminary data. Starting with a proof-of-concept trial in a single country using national funding would allow operationalizing the MASH IRP on a local scale. This approach would strengthen the CRN, provide insights into risks and benefits, and pave the way for future expansion to multiple countries. Successful proof-of-concept would potentially attract funding from small biotech companies and foster the establishment of a consortium involving private companies and academic leader across different countries[9,10,19].

Another key ingredient that might prove critical in order to succeed in the implementation of a MASH IRP is the engagement of investigators and patients and their communities. The survey results indicate a significant interest in participating in an IRP by a number of renowned and experienced MASH investigators. However, both in the scenario of an international, industry-funded trial for non-cirrhotic population entailing a large sample size, and for the case of a smaller academically-initiated PT in cirrhotic patients, a commitment to identify and recruit participants, and conduct the pre-established procedures with high-quality standards is of great importance.

## What type of CRN do we need?

One crucial aspect to consider when determining the type of CRN required to set up the MASH IRP is the assets that are available for leveraging. These assets may include previous research collaborations, established relationships with key stakeholders, access to patient databases, and existing infrastructure. By identifying and utilizing these assets effectively, we can enhance the efficiency and success of our clinical research efforts.

Clinical and research networks focused on MASH have been established to enhance research and knowledge in the field during the last decade. While there is currently no dedicated CRN specifically for operationalizing a MASH PT, various initiatives that might serve as inspiration and even support the future IRP already exist. The LITMUS[14] and NIMBLE[24] projects have developed CRNs to evaluate non-invasive biomarkers for MASH diagnosis and prognosis. Patient registries, such as the European NAFLD

Registry[15], currently recruiting in 13 countries, and the national HEPAmet registry in Spain, contribute real-world data on disease epidemiology and management[29,30]. The NIDDK NASH CRN in the USA have made significant contributions to defining histological criteria and conducting clinical trials[31–35]. The European Health Data and Evidence Network (EHDEN)[36], which is not a CRN, aims to create a federated data network for standardized access to EU citizens' data. Leveraging electronic health records to optimize the identification of potential candidates to the MASH IRP in the participating sites, or harmonizing queries, randomization and follow-up procedures, and incorporating clinical and complementary tests data to the platform clinical research data capture programs amongst other could greatly enhance efficiency, which is paramount in a logistically complex trial as are IRP.

The results of the survey should be carefully interpreted due to a series of limitations. The list of investigators to whom we reached out for the survey was built on sources that led to a selection bias (e.g., centers participating in ongoing MASH clinical trials, European NAFLD Registry recruiting centers, known investigators to EU-PEARL taskforce). In addition, the number of respondents is relatively low and does not represent the clinical research community in European sites as a whole. Likely, the number of Spanish sites that agreed to participate in the survey was larger compared to other countries because the reach out was done from a Spanish site. Moreover, although American and Asian investigators and industry partners acted as consultants to the EU-PEARL MASH team during the project, non-European stakeholders were not reached to participate in the survey. This clearly limits the breath of the conclusions regarding a potential global IRP. This was an exploratory survey in the context of a European project and therefore, in the future, this survey should be repeated in a more systematic manner by the CRO responsible for the logistics of the MASH IRP.

## Conclusion

The setting up of a PT in MASH requires navigating regulatory challenges, addressing patient recruitment barriers, securing funding, and fostering collaboration among academia, industry, and patient organizations. The determinants shaping the way forward include the regulatory landscape, population of interest, proof-of-concept considerations, and funding models. By addressing these factors and leveraging existing CRNs and initiatives, the establishment of a PT in MASH can advance research and accelerate the development of effective treatments for such a pressing public health concern characterized by its therapeutic gaps.

## Data availability

Anonymized individual responses to the survey are available upon request. Source data for Fig. 3 are available as Supplementary Data 3.

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

## Acknowledgements

We would like to express our gratitude to the individuals who contributed to the development and refinement of the survey used in this study. Special thanks go to Anna Sanjuan and Santiago Pérez-Hoyos (Vall d'Hebron Institute of Research). We would also like to extend our appreciation to Clara Sabiote (VHIR) for her meticulous proofreading of the survey, ensuring its clarity and coherence. Furthermore, we would like to acknowledge the supports and collaboration of the EU-PEARL MASH taskforce members, whose guidance and expertise were invaluable throughout the entire process. Their commitment to advancing research in MASH and their contributions to the formulation of the survey structure and content were essential in conducting this study. Lastly, we are grateful to all the respondents who took the time to respond to the survey and share their valuable insights and perspectives. Their contributions have provided valuable data for our analysis and have significantly enriched our understanding of the landscape surrounding the establishment of a PT in MASH. The names of those who agreed to it have been included in the list of MASH EU-PEARL collaborators below. EU-PEARL has received funding from the Innovative Medicines Initiative 2 Joint Undertaking under grant agreement No 853966- 2. This Joint Undertaking receives support from the European Union's Horizon 2020 Research and Innovation Programme and EFPIA and CHILDREN'S Tumor Foundation, Global Alliance for TB Drug Development Non-Profit Organisation, Springworks Therapeutics Inc. This publication reflects only the author's views. The JU is not responsible for any use that may be made of the information it contains. QMA is an NIHR Senior Investigator and is supported by the Newcastle NIHR Biomedical Research Centre. Partial results from this report were presented at Paris NASH Meeting 2022, NAFLD Summit 2022 and Innovations in NAFLD Care Workshop 2023.

## Author contributions

Conceptualization: E.S., J.M.P.; Methodology: E.S., J.M.P.; Data analysis: E.S., J.M.P.; Manuscript Writing: E.S., S.M.M., J.R.E., A.J., M.M.G., J.M.P.; Reviewing and editing: E.S., F.T., Q.M.A., N.D.P., M.S.K., S.M.M., J.R.E., A.J., M.M.G., F.K., J.G., V.R., J.M.P.; Approval of final draft: E.S., F.T., Q.M.A., N.D.P., M.S.K., S.M.M., J.R.E., A.J., M.M.G., F.K., J.G., V.R., J.M.P.; Supervision: J.M.P.; Project Administration. E.S

## Competing interests

The authors declare the following competing interests: **FT** lab' work has been supported by the German Research Foundation (DFG, Ta434/8-1, CRC/TR 362) and research grants from Gilead, Allergan, Bristol-Myers Squibb and Inventiva. **QMA** is coordinator of the EU IMI-2 LITMUS consortium, which is funded by the EU Horizon 2020 programme and EFPIA. This multistakeholder consortium includes industry partners. QMA has received research grant funding from AstraZeneca, Boehringer Ingelheim, and Intercept Pharmaceuticals, Inc.; has served as a consultant on behalf of Newcastle University for Alimentiv, Akero, AstraZeneca, Axcella, 89bio, Boehringer Ingelheim, Bristol Myers Squibb, Galmed, Genfit, Genentech, Gilead, GSK, Hanmi, HistoIndex, Intercept Pharmaceuticals, Inc., Inventiva, Ionis, IQVIA, Janssen, Madrigal, Medpace, Merck, NGM Bio, Novartis, Novo Nordisk, PathAI, Pfizer, Poxel, Resolution Therapeutics, Roche, Ridgeline Therapeutics, RTI, Shionogi, and Terns; has served as a speaker for Fishawack, Integritas Communications, Kenes, Novo Nordisk, Madrigal, Medscape, and Springer Healthcare; and receives royalties from Elsevier Ltd. **NAdP** works for Janssen. **MSK** is an employee of and a shareholder in Novo Nordisk A/S. **JRE** has received speaking fees from Gilead. **JG** has received consulting fees from Boehringer- Ingelheim, speaking fees from Echosens and travel expenses from Gilead and Abbie. Funds from ISCIII PI18/00947 and PI21/00691. **VR** consults for and Intercept, Novo Nordisk, Galmed, Poxel, NGM, Madrigal, Enyo, Sagimet, 89 Bio, Prosciento, Terns, and Theratechnologies, and received grants from Intercept and Gilead. **JMP** reports having received consulting fees from Boehringer-Ingelheim, MSD and Novo Nordisk. He has received speaking fees from Gilead, Intercept, and Novo Nordisk, and travel expenses from Gilead, Rubió, Pfizer, Astellas, MSD, CUBICIN, and Novo Nordisk. He has received educational and research support from Madrigal, Gilead, Pfizer, Astellas, Accelerate, Novartis, Abbvie, ViiV, and MSD. Funds from European Commission/EFPIA IMI2 853966-2, IMI2 777377, H2020 847989, and ISCIII PI19/01898.
