## [Peer Review File · Communications Medicine]

Reviewers' comments:

Reviewer #1 (Remarks to the Author):

The manuscript titled "Setting up a clinical research network for platform trials in metabolic-dysfunction associated-steatohepatitis" presents a comprehensive and detailed analysis of the steps and challenges involved in setting up an IRP for MASH. The study's emphasis on adaptive-design trials and the creation of a CRN is timely and relevant given the growing prevalence of MASH and the lack of effective treatments. The evaluated items are well chosen and assess the primary factors regarding future research in MASH. However, there are critical areas, particularly concerning research site selection and representation across the EU, that warrant further clarification and potential reconsideration.

- The basis for identifying specific research sites remains unclear. Could the authors delineate the criteria utilized for this selection? Were these criteria standardized across all countries? The transparency in this aspect is vital for understanding the study's representativeness.

- The manuscript notes an uneven distribution of invitations across EU countries, with some not receiving any. This raises questions about the inclusiveness of the study. Could the authors provide insights into why certain countries were underrepresented or omitted?

- It is unclear whether the invitations were directed to specific individuals or institutions. A detailed explanation of this selection process and the rationale for limiting invitations in certain countries, despite their significant population size and number of relevant centers, is essential.

- A more inclusive approach might have been to engage representatives from each country to nominate potential sites. This could have ensured more comprehensive coverage and awareness of eligible centers within each country.

- To enhance representativeness, why were national hepatology societies not consulted for potential site nominations? Their involvement could have provided a more exhaustive and representative list of centers, potentially improving the response rate and the study's overall inclusivity.

- It appears that certain well-qualified centers, including several in my country of practice, were overlooked despite their active involvement in ongoing MASH trials. This oversight suggests that the survey results might not fully capture the EU-wide perspective on MASH research.

- Please add details regarding the statistical analyses in the methods section.

- The higher response rates from countries like Spain and Italy correlate with the higher number of invitations they received. Actually, the highest acceptance rate was in Sweden, where all three invited sites participated.

- For sites without prior involvement in national or international MASLD or MASH registries, what were the inclusion criteria? The manuscript would benefit from a clearer explanation of why these sites were chosen over others with established research backgrounds in this area.

- Minor typo in the abstract: Methods: "potentially" instead of "potential".

To summarize, the current approach to site selection and survey dissemination appears to have limitations in terms of geographic coverage and representativeness. A reassessment of this process, possibly involving national societies, could enhance the breadth and validity of future research. Despite these concerns, the questions evaluated in the study lay a solid foundation for future research. However, addressing the aforementioned methodological concerns is crucial for ensuring broader applicability and relevance of the findings.

Reviewer #2 (Remarks to the Author):

Many thanks for the opportunity to review this manuscript.

Overall, the authors are to be commended on the attempting to present a contemporaneous representation of the field as it exists within the clinical landscape in relation to developing CRNs in PTs in MASH. The authorship consists of a number of leading authorities on trials in MASH and are well placed to undertake this assessment.

I do have a number of comments to be considered.

These mainly pertain to the methodology in which sites were selected. I appreciate the authors have discussed this as a limitation within the paper already, however, I feel it warrants further discussion, as well as some commentary about how this shortcoming may be overcoming in understanding a wider pan-European assessment?

Additionally, how generalizable are these findings more globally?

Reviewer #3 (Remarks to the Author):

This submitted manuscript by Sena et al., on behalf of the "EU-PEARL Investigators" describes data obtained from European clinical research sites on their interest in creating platform trials to study MASH (formerly nonalcoholic steatohepatitis), along with the possible incentives and concerns in undertaking the venture. The undertaking, if it comes to be, will be novel and likely important given the lack of currently labelled drugs to treat this highly prevalent disease/condition. The authors sent a questionnaire (provided in the supplemental material) to 146 potential investigators based on those investigators prior publication history on fatty liver, their participation in trials of such, or their being recommended by others. Unfortunately, there was a low response rate (N=57), indicating a lack of interest for most in participation. My comments and suggestions are as follows:

- 1) The title is rather colloquial. "Setting up" should be removed and the Clinical Research Network is somewhat redundant. I suggest the title to be: "Needs Assessment for Creation of a Platform Trial Network in Metabolic-dysfunction Associated Steatohepatitis".
- 2) The "Keywords" should have the full name of NASH and NAFLD
- 3) Abstract: the first sentence claims that clinical trial designs are responsible for the unmet need for effective drug treatment. I dispute this, as many drug trials thus far failed because the drug was not sufficiently effective. In the abstract Methods, there are numerous grammatical mistakes. In the abstract Conclusions, I do not believe the finding of 40 of 146 investigators having general interest "encouraging", given that most of the 40 require site funding by industry to engage, while most reported moderate or extreme difficulty in patient recruitment.
- 4) Introduction: Last sentence of paragraph 2, should include "standardization of protocols and interpretation" as well. Paragraph 3 line 96 should include common cores for radiology, pathology and statistics for centers, along with protocols for sample handling, distribution and management.
- 5) Methods: Table in supplement could be improved by listing each site with what manner they were selected for survey (publications versus European NAFLD registry vs secondary sources or multiple) and what technical capabilities in procedures they have that qualify them for the data needed on recruited subjects.

6) Results: Concerning that only 3 of the 53 respondents felt that platform trials offered extra benefits to standalone trials (line 178). This begs the question as to why the other 51 sites would be interested in PT.

7) Discussion: In multiple instances, the term "advanced fibrosis" is used without clarification of what stage that means (?F2, or only F3 or F4?). I believe it worthwhile to include discussion of what Committees would be formed, the governance structure, how sample allotment, publication credit, and idea generation priority is determined. Whether pediatric subjects will be included, and if not, why not?

8) References: #10 and #17 are the same.

9) Table 1: Listing "high screening failure rates" in table is not an unmet need. It should be "means to minimize screening failures". 10) Table 2: Lack of funding by drug owners isn't a challenge. Obtaining funding from drug owners is the challenge (i.e. avoid the double negative).

11) Figure 1A does not appear original. 1B could be in supplemental material.

12) Supp material Q3: On site experience needs a table with each question delineating the experience of each site

13) Appendix 2: along with "preselected sites" indicate which sites answered, yes or no.

Reviewer #4 (Remarks to the Author):

This paper reports the results of a European survey of hospitals that have conducted clinical research in NASH to learn whether there is an appetite for a platform trial.

Some of the data is interesting especially concerning the investigators' preference for non-invasive testing rather than liver biopsy. Although this is perhaps more pertinent to discussions with the regulatory bodies than this scenario. It was also interesting to learn that physicians believed that their priority would be those with advanced fibrosis/cirrhosis, which, given there are no effective anti-fibrotic drugs might not be in line with the views of the companies developing these drugs.

I have a few points to make.

In the abstract I do not necessarily agree with the following statements

1. "Traditional clinical trial designs for drug development in metabolic-dysfunction associated steatohepatitis (MASH), formerly known as non-alcoholic steatohepatitis (NASH), have several limitations, resulting in an unmet need for effective drug treatments." Surely the unmet need exists because no effective drug treatments have been developed. It seems unlikely that a more

innovative trial design would change the fact that the drugs so far haven't worked (with the potential exception of Madrigal's drug).

2. "as investigators have shown a general interest in actively participating, either in its setup and/or as recruiting sites." As only 39% of the investigators responded.

My major criticism is that the questionnaire did not include sufficient detail as to what such a platform for NASH might require for each site. The STAMPEDE platform trial in the UK ran from 2005-2023 and recruited 12000 men with Prostate cancer. These trials are huge undertakings. Perhaps the authors could have presented some "dummy" sample size calculations and theoretical trials arms to enable respondents to assess whether such an enterprise was remotely feasible. I note the respondents would prefer to study those with NASH and advanced fibrosis which may only account for 5-10% of NAFLD/MASLD patients (depending on clinical characteristics). Presenting sites with required numbers of patients to recruit and likely time frame to do this and asking whether they had sufficient resources to run such a trial would provide valuable data, in particular to aid discussions with potential funders, which I assume the authors will do. Adding an estimated likely budget for this would also have been very interesting and help assess whether this endeavour (which is very exciting) would be remotely feasible.

Reviewer #1

1. The manuscript titled "Setting up a clinical research network for platform trials in metabolic-dysfunction associated-steatohepatitis" presents a comprehensive and detailed analysis of the steps and challenges involved in setting up an IRP for MASH. The study's emphasis on adaptive-design trials and the creation of a CRN is timely and relevant given the growing prevalence of MASH and the lack of effective treatments. The evaluated items are well chosen and assess the primary factors regarding future research in MASH. However, there are critical areas, particularly concerning research site selection and representation across the EU, that warrant further clarification and potential reconsideration.

A: We appreciate the time devoted by the reviewer to revise and provide meaningful input to improve our manuscript.

2. The basis for identifying specific research sites remains unclear. Could the authors delineate the criteria utilized for this selection? Were these criteria standardized across all countries? The transparency in this aspect is vital for understanding the study's representativeness.

A: Thank you for the observation. The selection process is described in the Methods section "Identification of sites" in page 4 of the revised version of the manuscript. Sites were identified by using public data on a) clinicaltrials.gov b) centers associated with the NAFLD registry/other national registries c) recommendations. Specifically:

- a) Active sites in clinical trials related to MASH were identified.
- b) We contacted sites that previously participated in registries as the NAFLD registry, MALFD/MASH registries and the Hepamet registry in Spain.
- c) The survey in this study included an option for the site to recommend other researchers who might be interested to participate in the project.

The limitations in the site selection are acknowledged in the Discussion (see page 9 of the manuscript). Even though we might had not reached some sites, the participation rate in this study shows the difficulty of the international multicentric studies, evidencing the limitations to engage with centers although a direct invitation. We are cognizant that when the time of setting up a CRN for a MASH PT arrives we will need to expand the survey and adapt it to include more sites and countries.

2. The manuscript notes an uneven distribution of invitations across EU countries, with some not receiving any. This raises questions about the inclusiveness of the study. Could the authors provide insights into why certain countries were underrepresented or omitted?

A: The reviewer is right. In spite of our, in theory, comprehensive approach to select sites to be invited to the survey, there are three biases that largely conditioned our results. The first one is conscious, namely only inviting European sites because of the scope of the EU-PEARL project. However, using sites mostly participating in the NAFLD registry and clinical trials created a reference bias, whereas using Hepamet as a more local contact hub also generated a bias towards Spanish sites. In addition, the investigators involved in the EU-PEARL project were not invited to answer the survey. This is acknowledged in the limitations and will need

further iterations with a broader scope and stronger selection criteria when the IRP is eventually set up. Thanks.

3. It is unclear whether the invitations were directed to specific individuals or institutions. A detailed explanation of this selection process and the rationale for limiting invitations in certain countries, despite their significant population size and number of relevant centers, is essential.

We appreciate the remark. We confirm that each invitation described in the manuscript (Figure 2) corresponds to sites and not to researchers of the same site. The methods section describes the selection process (see page 4 of the revised version of the manuscript, Methods section), which was inclusive, trying to reach to every researcher found to be involved in MASH clinical research, there were no exclusive criteria. The limitation had been exposed in the manuscript, being an important the missing of certain contact information to send the invite to participate in the study.

4. A more inclusive approach might have been to engage representatives from each country to nominate potential sites. This could have ensured more comprehensive coverage and awareness of eligible centers within each country.

Thank you for the comment. We agree that the designation of a specific representative by country would help in the identification of potential sites. However, the methods of the study were as described, and the limitations had been exposed. The results will help in the design of future survey studies to amplify the participation and reach to more sites. It is worth mentioning that the survey included a specific question that encouraged participants to nominate other colleagues that might be interested in the survey in a way to reach the most sites as possible.

5. To enhance representativeness, why were national hepatology societies not consulted for potential site nominations? Their involvement could have provided a more exhaustive and representative list of centers, potentially improving the response rate and the study's overall inclusivity.

We considered sites recognized as experienced with the capability to recruit MASH patients, which is generally well represented in the sources we relied on (i.e., mostly lists of participants in MASH clinical trials). Consulting national hepatology societies might be a good strategy to consider in future survey studies.

6. It appears that certain well-qualified centers, including several in my country of practice, were overlooked despite their active involvement in ongoing MASH trials. This oversight suggests that the survey results might not fully capture the EU-wide perspective on MASH research.

Thank you for the comment. The study captures an EU-wide perspective on MASH research by an international multicentric representation, we managed to get the participation of 15 different countries and reflects a great organizational effort. There is an intrinsic complexity to get all the EU countries to participate (even if invited, as there were no responses by some sites). Survey studies usually have this limitation and we managed to deliver objective data using the methods displayed. Every study can be improved for future and detailing the limitations of our study will also help future researchers to consider important aspects for

designing survey studies. In our study we attempted to present a contemporaneous representation of the field as it exists within the clinical landscape including quite few key opinion leaders on MASLD/MASH research.

7. Please add details regarding the statistical analyses in the methods section.

Thank you for the suggestion. You may find the statistical analysis description in **page 5** of the revised version of the manuscript.

The answers to the survey were expressed as absolute frequencies and percentages (%). The survey was developed and performed using the REDCap electronic data capture tools. Raw data and results were directly extracted from the platform.

8. The higher response rates from countries like Spain and Italy correlate with the higher number of invitations they received. Actually, the highest acceptance rate was in Sweden, where all three invited sites participated.

As mentioned in the response to comment 2, in the case of Spain it is likely that there is a response bias because the coordinators of the EU-PEARL project and the work package leads tackling MASH were from a Spanish institution. For instance, participants in the Hepamet cohort disproportionately accepted to participate in the survey compared to those in the International NAFLD Registry overall. However, we would like to emphasize that we sent invitations to all sites included in the listed generated from the sources mentioned above.

9. For sites without prior involvement in national or international MASLD or MASH registries, what were the inclusion criteria? The manuscript would benefit from a clearer explanation of why these sites were chosen over others with established research backgrounds in this area.

For those sites not included in MASLD or MASH registries, we included them based on their activity in clinical trials or clinical research in MASLD/MASH.

10. Minor typo in the abstract: Methods: "potentially" instead of "potential".

Thanks, corrected.

To summarize, the current approach to site selection and survey dissemination appears to have limitations in terms of geographic coverage and representativeness. A reassessment of this process, possibly involving national societies, could enhance the breadth and validity of future research. Despite these concerns, the questions evaluated in the study lay a solid foundation for future research. However, addressing the aforementioned methodological concerns is crucial for ensuring broader applicability and relevance of the findings.

Thanks again for your valuable insight.

Reviewer #2

1. Many thanks for the opportunity to review this manuscript. Overall, the authors are to be commended on the attempting to present a contemporaneous representation of the field as it exists within the clinical landscape in relation to developing CRNs in PTs

in MASH. The authorship consists of a number of leading authorities on trials in MASH and are well placed to undertake this assessment.

Thank you for your time and interest in our work.

2. I do have a number of comments to be considered. These mainly pertain to the methodology in which sites were selected. I appreciate the authors have discussed this as a limitation within the paper already, however, I feel it warrants further discussion, as well as some commentary about how this shortcoming may be overcoming in understanding a wider pan-European assessment?

We appreciate the reviewer's point. The survey was meant as an assessment for the future set up of the platform trial in MASH in some European sites. We acknowledge that some countries are underrepresented, but we believe the answers represent widely opinions of key opinion leaders on MAFLD/MASH research. We have tried to strengthen the explanations surrounding the methods on site selection, invitations and responses obtained, also following the comments by Reviewer 1.

3. Additionally, how generalizable are these findings more globally?

The reviewer is right that our findings might not be generalizable outside the UE. For instance, North American sites might not approach clinical research and the drug development pipeline in a similar manner. At first, we planned to send the survey to centers in South America and the United States, but for the feasibility of setting an academic platform trial we focused on European sites. Therefore, if the platform trial were to be set up in Europe and operationally worked out, we might be interested in exploring the interest of sites outside Europe.

Reviewer #3:

This submitted manuscript by Sena et al., on behalf of the "EU-PEARL Investigators" describes data obtained from European clinical research sites on their interest in creating platform trials to study MASH (formerly nonalcoholic steatohepatitis), along with the possible incentives and concerns in undertaking the venture. The undertaking, if it comes to be, will be novel and likely important given the lack of currently labelled drugs to treat this highly prevalent disease/condition. The authors sent a questionnaire (provided in the supplemental material) to 146 potential investigators based on those investigators prior publication history on fatty liver, their participation in trials of such, or their being recommended by others. Unfortunately, there was a low response rate (N=57), indicating a lack of interest for most in participation.

We appreciate the time and interest devoted to revise our manuscript.

1) The title is rather colloquial. "Setting up" should be removed and the Clinical Research Network is somewhat redundant. I suggest the title to be: "Needs Assessment for Creation of a Platform Trial Network in Metabolic-dysfunction Associated Steatohepatitis".

We agree with the reviewer's assessment and have changed it following his/her suggestion.

2) The "Keywords" should have the full name of NASH and NAFLD

The reviewer is right. We have rewritten them accordingly.

3) Abstract: the first sentence claims that clinical trial designs are responsible for the unmet need for effective drug treatment. I dispute this, as many drug trials thus far failed because the drug was not sufficiently effective. In the abstract Methods, there are numerous grammatical mistakes. In the abstract Conclusions, I do not believe the finding of 40 of 146 investigators having general interest "encouraging", given that most of the 40 require site funding by industry to engage, while most reported moderate or extreme difficulty in patient recruitment.

The reviewer has a point here, no doubt. We have modified the text accordingly.

4) Introduction: Last sentence of paragraph 2, should include "standardization of protocols and interpretation" as well. Paragraph 3 line 96 should include common cores for radiology, pathology and statistics for centers, along with protocols for sample handling, distribution and management.

Thank you for the observation. You may find the changes in the Introduction section on page 3 of the revised version of the manuscript.

5) Methods: Table in supplement could be improved by listing each site with what manner they were selected for survey (publications versus European NAFLD registry vs secondary sources or multiple) and what technical capabilities in procedures they have that qualify them for the data needed on recruited subjects.

We have modified Supplementary Table 1 and added a Supplementary Table 2 to describe the selection criteria for each site. As for the qualification to recruit subjects, we assumed that participating in either the international NAFLD registry or having been selected to participate in international clinical trials on MASH capacitated them to recruit patients.

6) Results: Concerning that only 3 of the 53 respondents felt that platform trials offered extra benefits to standalone trials (line 178). This begs the question as to why the other 51 sites would be interested in PT.

Thank you for the comment. The results of Figure 3D were poorly summarized in the text. sentence should read "None of the respondents stated that PT offer no extra benefits to patients compared to traditional standalone trials, only 6% of respondents believed that PT offer *few* extra benefits, whereas the rest of the respondents considered that PT provide either moderate or many benefits to patients". The text has been modified accordingly.

7) Discussion: In multiple instances, the term "advanced fibrosis" is used without clarification of what stage that means (?F2, or only F3 or F4?). I believe it worthwhile to include discussion of what Committees would be formed, the governance structure, how sample allotment, publication credit, and idea generation priority is determined. Whether pediatric subjects will be included, and if not, why not?

Thank you for the observation. We have now included a straightforward definition on page 6 of the revised version of the manuscript. Regarding governance, this was addressed in a specific separate deliverable of the EU-PEARL project (D6.7). However, we have added some information on the discussion regarding this question and referenced such deliverable. Regarding pediatric patients, they would not be included in the initial platform because the

whole idea of the platform is to have a shared placebo arm and a common cohort to screen patients to increase efficiency. Pediatric patients in MASH have a completely different and separate pipeline with largely distinct design, duration, etc.

8) References: #10 and #17 are the same.

Fixed, thanks

9) Table 1: Listing "high screening failure rates" in table is not an unmet need. It should be "means to minimize screening failures".

Modified accordingly.

10) Table 2: Lack of funding by drug owners isn't a challenge. Obtaining funding from drug owners is the challenge (i.e. avoid the double negative).

Modified accordingly.

11) Figure 1A does not appear original. 1B could be in supplemental material.

Figure 1 A was designed and drawn by the article' authors. We have moved Figure 1B to the Supplementary Material

12) Supp material Q3: On site experience needs a table with each question delineating the experience of each site

Thank you for the suggestion. Supplementary Table 2 has been created to describe the experience in clinical research of the participating sites.

13) Appendix 2: along with "preselected sites" indicate which sites answered, yes or no.

Done as suggested, thanks.

Reviewer #4

1. This paper reports the results of a European survey of hospitals that have conducted clinical research in NASH to learn whether there is an appetite for a platform trial. Some of the data is interesting especially concerning the investigators' preference for non-invasive testing rather than liver biopsy. Although this is perhaps more pertinent to discussions with the regulatory bodies than this scenario. It was also interesting to learn that physicians believed that their priority would be those with advanced fibrosis/cirrhosis, which, given there are no effective anti-fibrotic drugs might not be in line with the views of the companies developing these drugs.

Thanks for your time and through revision of our work.

2. In the abstract I do not necessarily agree with the following statements "Traditional clinical trial designs for drug development in metabolic-dysfunction associated steatohepatitis (MASH), formerly known as non-alcoholic steatohepatitis (NASH), have several limitations, resulting in an unmet need for effective drug treatments". Surely the unmet need exists because no effective drug treatments have been developed. It seems unlikely that a more innovative trial design would change the fact that the drugs so far haven't worked (with the potential exception of Madrigal's drug).

The reviewer is right. We have modified the sentence accordingly.

3. "as investigators have shown a general interest in actively participating, either in its setup and/or as recruiting sites." As only 39% of the investigators responded.

The reviewer is right, we have rephrased the sentence to clarify that we referred to respondents of the survey.

4. My major criticism is that the questionnaire did not include sufficient detail as to what such a platform for NASH might require for each site. The STAMPEDE platform trial in the UK ran from 2005-2023 and recruited 12000 men with Prostate cancer. These trials are huge undertakings. Perhaps the authors could have presented some "dummy" sample size calculations and theoretical trials arms to enable respondents to assess whether such an enterprise was remotely feasible.

Thank you for the observation, which is very interesting. We have cited the documents where the statistical considerations to designing the master protocol for the platform trials were drawn from.

5. I note the respondents would prefer to study those with NASH and advanced fibrosis which may only account for 5-10% of NAFLD/MASLD patients (depending on clinical characteristics). Presenting sites with required numbers of patients to recruit and likely time frame to do this and asking whether they had sufficient resources to run such a trial would provide valuable data, in particular to aid discussions with potential funders, which I assume the authors will do.

Indeed. As part of EU-PEARL, such "feasibility questionnaires" were sent to a sample of sites. This step will be absolutely crucial before launching a potential future PT on MASH. Thanks.

6. Adding an estimated likely budget for this would also have been very interesting and help assess whether this endeavour (which is very exciting) would be remotely feasible.

Thanks for the suggestion. In line with the response to comment 4 and to the reply to comment 7 of reviewer 3, these aspects were addressed elsewhere during the EU-PEARL project. We have added a short comment in the discussion as well as appropriate references.

REVIEWERS' COMMENTS:

Reviewer #1 (Remarks to the Author):

The manuscript has been modified according to the previously mentioned comments. I believe the manuscript has been well modified and can be considered for publication.

Reviewer #2 (Remarks to the Author):

Many thanks for the significant efforts in amending the relevant points raised, I think the manuscript is enhanced accordingly.

Reviewer #3 (Remarks to the Author):

The authors have made commendable and appropriate revisions to the reviewers contents, addressing all that could be revisited and addressed. I find this version ready for publication.

Joel E. Lavine, MDPHD

Professor of Pediatrics

Columbia University (New York)

Reviewer #4 (Remarks to the Author):

The authors have addressed the points that I raised. I sincerely wish them all the very best with this endeavour.

Alastair O'Brien